# FedPD: Defying data heterogeneity through privacy distillation

## Abstract

Model performance of federated learning (FL) typically suffers from data heterogeneity, i.e., data distribution varies with clients. Advanced works have already shown great potential for sharing client information to mitigate data heterogeneity. Yet, some literature shows a dilemma in preserving strong privacy and promoting model performance simultaneously. Revisiting the purpose of sharing information motivates us to raise the fundamental questions: Which part of the data is more critical for model generalization? Which part of the data is more privacy-sensitive? Can we solve this dilemma by sharing useful (for generalization) features and maintaining more sensitive data locally? Our work sheds light on data-dominated sharing and training, in a way that we decouple original training data into sensitive features and generalizable features. To be specific, we propose a **Fed**erated **P**rivacy **D**istillation framework named FedPD to alleviate the privacy-performance dilemma. Namely, FedPD keeps the distilled sensitive features locally and constructs a global dataset using shared generalizable features in a differentially private manner. Accordingly, clients can perform local training on both the local and securely shared data for acquiring high model performance and avoiding the leakage of not distilled privacy. Theoretically, we demonstrate the superiority of the sharing-only useful feature strategy over sharing raw data. Empirically, we show the efficacy of FedPD in promoting performance with comprehensive experiments.

## 1 Introduction

Federated learning (FL), as an emerging protection paradigm, receives increasing attention recently (Kairouz et al., 2021; Li et al., 2021b; Yang et al., 2019), which preserves data privacy without transmitting pure data. In general, distributed clients collaboratively train a global model by aggregating gradients (or model parameters). However, distributed data can cause heterogeneity issues (McMahan et al., 2017; Li et al., 2022; 2020; Zhao et al., 2018), due to diverse computing capability and non-IID data distribution across federated clients. It results in unstable convergence and degraded performance.

To address the challenge of heterogeneity, the seminal work, federated averaging (FedAvg) (McMahan et al., 2017), proposes weighted averaging to overcome Non-IID data distribution when sharing selected local parameters in each communication round. Despite addressing the diversity of computing and communication, FedAvg still struggles with the client drift issue (Karimireddy et al., 2020). Therefore, recent works try to resolve this issue by devising new learning objectives (Li et al., 2020), designing new aggregation strategies (Yurochkin et al., 2019) and constructing information for sharing (Zhao et al., 2018; Yoon et al., 2021). Among these explorations, sharing relevant information across clients provides a straightforward and promising approach to mitigate data heterogeneity.

However, recent works point out a dilemma in preserving strong privacy and promoting model performance. Specifically, (Zhao et al., 2018) show that a limited amount of sharing data could significantly improve training performance. Unfortunately, sharing raw data, synthesized data, logits and statistical information (Luo et al., 2021; Goetz & Tewari, 2020; Hao et al., 2021; Karimireddy et al., 2020) can incur high privacy risks. To protect clients' privacy, differential privacy (DP) provides a de facto standard way for provable security quantitatively. The primary concern in applying DP is about performance degradation (Tramer & Boneh, 2020). Thus, solving the above dilemma can contribute to promoting model performance while preserving strong privacy.

## 1.1 Systematic Overview of FedPD

To solve the dilemma, we revisit the purpose of sharing information: sharing raw data benefits model generalization while violating privacy leakage. This motivates us to raise the fundamental questions:

*(1) Is it necessary to share complete raw data features to mitigate data heterogeneity?*
We find that some data features are more important than others to train a global model. Therefore, an intuitive approach is to divide the data features into two parts: one part for model generalization, named generalizable features, and the other part with clients' privacy, named sensitive features. Then, the dilemma can be solved by sharing generalizable features and keeping sensitive features locally throughout the training procedure. The insight is that the sensitive features in the data are kept locally, and the generalizable features intrinsically related to generalization are shared across clients. Accordingly, numerous decentralized clients can share generalizable features without privacy concerns and construct a global dataset to perform local training.

*(2) How to divide data features into generalizable features and sensitive features?*
It is challenging to identify which part of the data is more important for model generalization and which part is more privacy-sensitive. To resolve this challenge, we propose a novel framework named **Fed**erated **P**rivacy **D**istillation (FedPD). FedPD introduces a competitive mechanism by decomposing $x \in \mathbb{R}^d$ with dimension $d$ into generalizable features $x_g \in \mathbb{R}^d$ and sensitive features $x_s \in \mathbb{R}^d$, i.e., $x = x_g + x_s$. In FedPD, sensitive features $x_s$ aim to cover almost all information in the data $x$, while the generalizable features $x_g$ compete with $x_s$ for extracting sufficient information to train models such that models trained on $x_g$ can generalize well. Consequently, the sensitive features are almost the same as the data while models trained on generalizable features generalize well.

*(3) What is the difference between sharing raw data features and partial features?*
To ensure that sharing the generalizable features $x_g$ cannot expose FL to the danger of privacy leakage, we follow the conventional style in applying differential privacy to protect generalizable features $x_g$ shared across clients. Our trick is that most information in data has been distilled as sensitive features $x_s$, which is very secure and kept locally. In other words, we only need a relatively small noise to protect $x_g$, without the need to fully protect the raw data $x$, yet achieving a much stronger privacy than the straightforward protection (i.e., directly sharing $x$ with differential privacy). Intuitively, sharing partial information in the data is more accessible to preserve privacy than sharing complete information, which is fortunately consistent with our theoretical analysis.

## 1.2 Our Results and Contribution

To tackle data heterogeneity, we propose a novel framework with privacy, which constructs a global dataset using securely shared data and performs local training on both the local and shared data, shedding new light on data-dominated sharing schemes. To show the efficacy, we deploy FedDP on four popular FL algorithms, including FedAvg, FedProx, SCAFFOLD, and FedNova, and conduct experiments on various scenarios with respect to different amounts of devices and varying degrees of heterogeneity. Our extensive results show that FedPD achieves considerable performance gains on different FL algorithms. Our solution not only improves model performance in FL but also provides strong security, which is theoretically guaranteed from the lens of differential privacy.

Our contributions are summarized as follows:

- We raise a foundation question: whether it is necessary to share complete raw data features when sharing privacy data for mitigating data heterogeneity in FL.
- We answer the question by proposing a plug-and-play framework named FedPD, where raw data features are divided into generalizable features and sensitive features. In FedPD, the sensitive features are distilled in a competitive manner and kept locally, while the generalizable features are shared in a differentially private manner to construct a global dataset.
- We give a new perspective on employing differential privacy that adds noise to partial data features instead of the complete raw data features, which is theoretically superior to the raw data sharing strategy.
- Extensive experiments demonstrate that FedPD can considerably improve the performance of FL models.

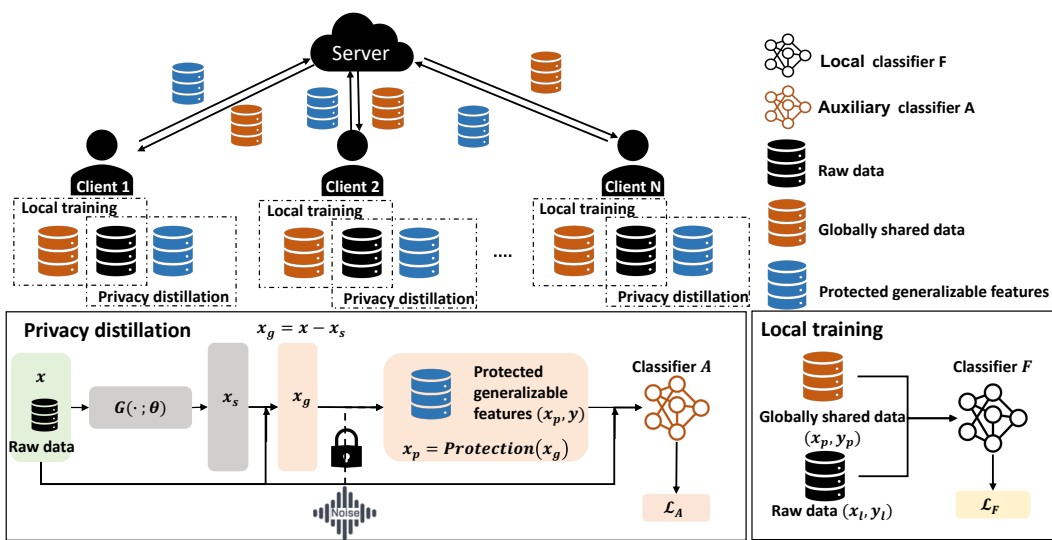

Figure 1: FL Framework with the plug-in FedPD. Clients generate generalizable features and add noise protection to get protected generalizable features $x_p$ during privacy distillation process. The protected generalizable features $x_p$ are collected from numerous distributed clients to construct a globally shared dataset while sensitive features $x_s$ are kept locally. During local training procedure, local raw data and a subset of globally shared data jointly train the local model for global aggregation. $\mathcal{L}_A$ denotes the Eq. 1 and $\mathcal{L}_F$ is the Eq. 3 in our paper.

## 2 RELATED WORK

**Federated Learning with heterogeneous data.** The classic FL algorithm FedAvg (McMahan et al., 2017) suffers from serious performance degradation when meeting severe Non-IID data. To address the data heterogeneity problem, a series of works propose a new learning objective to calibrate the updated direction of local training from being too far away from the global model, including FedProx(Li et al., 2020), FedIR (Hsu et al., 2020), SCAFFOLD (Karimireddy et al., 2020) and MOON (Li et al., 2021a). And some works propose designing new model aggregation schemes like FedAvgM (Hsu et al., 2019),FedNova (Wang et al., 2020b),FedMA (Wang et al., 2020a),FedBN (Li et al., 2021c).

Another promising direction is sharing some data, which mainly focuses on synthesizing and sharing data of different clients to mitigate client drift (Zhao et al., 2018; Jeong et al., 2018; Long et al., 2021). To avoid privacy leakage caused by sharing data, some methods share the statistics of data (Yoon et al., 2021; Shin et al., 2020), which still contains some raw data content. Some methods distribute intermediate features (Hao et al., 2021), logits (Chang et al., 2019; Luo et al., 2021), or the learned new embedding (Tan et al., 2022). Although these tactics enhance privacy at some degree, advanced attacks can still successfully reconstruct raw data given shared data (Zhao et al., 2020). Unlike prior research, we exploit DP to ensure privacy of shared data and then analyze privacy-performance trade-off.

**Differential privacy in federated learning**. Recent works on model memorization and gradient leakage confirm that model parameters are seemingly secure (Carlini et al., 2019). Training with differential privacy (Zhu et al., 2019; Nasr et al., 2019) is a feasible solution to avoid some attacks, albeit at some loss in utility. Differential privacy quantifies what extent individual privacy in a statistical dataset is preserved while releasing the established model over specific datasets.

In FL, training with differential privacy, i.e., adding noise to the model/data, originally aims to protect local information of each client (Yuan et al., 2019; Thakkar et al., 2019). Some works analyze the relation between convergence and utility in FL (Huang et al., 2020; Wei et al., 2020). A series of works in DP add noise to gradients or model parameters in FL to protect model privacy (Kim et al., 2021; van der Hoeven, 2019; Triastcyn & Faltings, 2019; Sun et al., 2021). Unlike model-based protection, our work aims to protect data privacy and mitigate the client drift issue. We provide a detailed discussion of exciting works in Appendix A.5.

# 3 METHODOLOGY AND DETAILED CONSTRUCTION

This section elaborates Federated Privacy Distillation (FedPD), which is illustrated in Figure 1. Our insight is to keep sensitive features on the client's side locally and share generalizable features globally in a differentially private manner. FedPD endows each client to use its local raw data features and generalizable features from others during local training, thus defying data heterogeneity.

## 3.1 DIVISION OF TWO TYPES OF FEATURES FOR PRIVATE DATA

Differential privacy (DP) is promising in FL protections, but sharing all raw data in a DP manner typically causes performance degradation. Recall that the goal of sharing information is to benefit the model generalization rather than to collect private information. Therefore, we suggest to share useful features (generalizable features) in data while keeping most features locally (sensitive features), such that shared features benefits global generalization and locally kept features avoids privacy leakage.

Ideally, if we could identify the sensitive features $\boldsymbol{x}_s$ and the generalizable features $\boldsymbol{x}_g$, we could be able to solve the privacy-performance dilemma. Intuitively, sensitive features $\boldsymbol{x}_s$ contain most information of data, while generalizable features $\boldsymbol{x}_g$ contains the nonsensitive part that can help global generalization in FL. To resolve the dilemma in protecting privacy and promoting performance, we can keep the sensitive features locally while sharing generalizable feature protected under differentially private guarantee. The major challenge here is that the intersection of two types of features as aforementioned may not be the empty set, making it challenging to distill privacy.

## 3.2 PRIVACY DISTILLATION

To address this issue, we propose a competitive mechanism to perform privacy distillation. Therein, the generalizable features aim to train models for generalizing well on the raw data, while the sensitive features compete with the generalizable features to construct the raw data. Consequently, the sensitive features is almost the same as the data while models trained on generalizable features generalizing well on the raw data. We propose two approaches to instantiate the competitive mechanism for privacy distillation, i.e., making generalizable features useful for model generalization while keeping sensitive features almost the same as the raw data, i.e., covering almost all information of raw data.

### 3.2.1 OPTIMIZATION VIEW

A straightforward approach is to distill private information in a meta manner (Finn et al., 2017). Specifically, we employ a generative model, e.g., a variational auto-encoder (VAE), $G(\cdot; \theta)$ parameterized with $\theta$ to achieve the goal of covering all information of raw data, i.e., $\boldsymbol{x}_s = G(\cdot; \theta)$ aims to reconstruct $\boldsymbol{x}$. Meanwhile, to ensure the generalizable features, $\boldsymbol{x}_g = \boldsymbol{x} - \boldsymbol{x}_s = \boldsymbol{x} - G(\cdot; \theta) = \boldsymbol{x}_g(\theta)$, useful for model generalization, we train an auxiliary classifier $A(\cdot; w)$ parameterized with $w$ using $\boldsymbol{x}_g$ such that $A(\cdot; w)$ trained on $\boldsymbol{x}_g$ performs well on the raw data $\boldsymbol{x}$. Then, we can formalize the task of privacy distillation into the following optimization problem as:

$$\min_{\theta} \mathbb{E}_{(\boldsymbol{x}, y)} \mathcal{L}(A(\boldsymbol{x}; \hat{w}(\theta)), y) + \mathcal{H}(\boldsymbol{x}_g(\theta)),$$
$$s.t. \ \hat{w}(\theta) = \arg\min_{w} \mathbb{E}_{(\boldsymbol{x}_g(\theta), y)} \mathcal{L}(A((\boldsymbol{x}_g(\theta); w), y), \boldsymbol{x}_g(\theta) = \boldsymbol{x} - G(\boldsymbol{x}; \theta). \tag{1}$$

Here, $y$ is the label of the sample $\boldsymbol{x}$ and the generalizable features $\boldsymbol{x}_g(\theta)$, $\hat{w}(\theta)$ is a function of $\theta$ denoting the parameters of classifier $A(\boldsymbol{x}; \cdot)$, $\mathcal{H}(\boldsymbol{x}_g(\theta))$ is the information entropy of $\boldsymbol{x}_g(\theta)$, and $\mathcal{L}(\cdot, \cdot)$ represents the cross-entropy loss. We can see that every possible parameter $\theta$ is paired with a model trained on the corresponding generated data $\boldsymbol{x}_g(\theta)$. Thus, solving the optimization problem is equivalent to searching for parameters $\theta$ to generate the generalizable features $\boldsymbol{x}_g(\theta)$ with minimum information entropy. Moreover, the model $A((\boldsymbol{x}_g(\theta); w), y)$ trained using $(\boldsymbol{x}_g(\theta), y)$ can perform well on the raw data.

However, the proposed non-convex optimization problem is non-trival. We employ a simple yet effective trick widely used in reinforcement learning (Mnih et al., 2015). Specifically, we alternatively update $G(\boldsymbol{x}; \theta)$ over $\boldsymbol{x}$ via stochastic gradient descent and update $A(\boldsymbol{x}_g(\theta); w)$ over $\boldsymbol{x}_g(\theta)$. Moreover, we minimize an upper bound of $\mathcal{H}(\boldsymbol{x}_g(\theta))$ with the variance of $\boldsymbol{x}_g(\theta)$ following (Ahuja et al., 2021).

### 3.2.2 GENERALIZATION VIEW

Besides the optimization approach, we also provide a generalization view to distill privacy. In a high level, we aim to train a model $A(\cdot; w)$ using $\boldsymbol{x}_g$ such that $A(\cdot; w)$ can generalize well on $\boldsymbol{x}$, i.e., samples drawn from a different distribution. Therefore, we should model how the performance on the generated data transfers to the raw data. To derive a detailed connect between these two distributions, the metric to measure the generalization performance should be defined clearly. According to the margin theory (Koltchinskii & Panchenko, 2002) that maximizing the margin between data points and the decision boundary achieves strong generalization performance, we relate such a margin to the generalization performance:

**Definition 3.1** (Margin). *We define the margin for a classifier $A(\cdot; w)$ on a distribution $\mathcal{P}$ with a distance metric $d$: $M_m(A, \mathcal{P}) = \mathbb{E}_{(\boldsymbol{x}, y) \sim \mathcal{P}} \inf_{A(\boldsymbol{x}') \neq y} d(\boldsymbol{x}', \boldsymbol{x})$.*

Built upon the defined margin that quantifies the degree of generalization performance, we can quantify the generalization performance of $A(\cdot; w)$ on a given distribution. To be specific, large margin means strong generalization performance.

A recent work (Tang et al., 2022) shows that the margin is intrinsically related to the distribution discrepancy in the representation space, i.e., the distance between distributions sampling $\boldsymbol{x}$ and that sampling $\boldsymbol{x}_g$. Thus, we propose minimizing the distribution discrepancy of the generated distribution and the raw distribution in the representation space:

$$\min_\theta \ \mathbb{E}_{(\boldsymbol{x}, y)} \mathcal{L}(A(\boldsymbol{x}; w), y) + \mathcal{L}(A(\boldsymbol{x}_g(\theta); w), y) + \mathcal{H}(\boldsymbol{x}_g(\theta)) + d(r(\boldsymbol{x}_g(\theta)), r(\boldsymbol{x})). \quad (2)$$

where $d$ is the distance metric used in the definition of margin and $r(\boldsymbol{x}_g(\theta))$ stands for the representation of $\boldsymbol{x}_g(\theta)$ generated by the classifier $A$.

### 3.3 DIFFERENTIALLY PRIVATE GENERALIZABLE FEATURES

The proposed privacy distillation methods make it possible to keep most (private) information locally while sending the generalizable features to the server. However, for ease of calculation of information entropy, we employ the variance of generalizable features as a surrogate, which may cause privacy leakage. This breaks the original intention of federated learning in protecting privacy. Thus, the shared generalizable features should be protected. Accordingly, the server can construct a global dataset using these generalizable features and send the dataset back to clients for local training.

To avoid privacy leakage, additional noise (e.g., Gaussian or Laplacian) is added to generalizable features $\boldsymbol{x}_g$, i.e., $\boldsymbol{x}_p \triangleq \boldsymbol{x}_g + \mathcal{N}(0, \sigma^2)$. Then, clients send $\boldsymbol{x}_p$ to the server to construct a globally shared dataset. Using the global dataset, clients can train classifier $F(\cdot; \phi)$ parameterized by $\phi$ with the local and shared data, :

$$\min_\phi \mathcal{L}_F(\phi) = \mathbb{E}_{(\boldsymbol{x}, y)} \mathcal{L}(F(\boldsymbol{x}; \phi), y) + \mathbb{E}_{(\boldsymbol{x}_p, y)} \mathcal{L}(F(\boldsymbol{x}_p; \phi), y). \quad (3)$$

Algorithm 1 summarizes the training procedure of FedAvg with FedPD. To make sure the framework can be used without privacy concern, we further provide the corresponding analysis. Before that, we introduce the definition of differential privacy, which we used for adding i.i.d noise to generalizable features.

**Definition 3.2.** *(Differential Privacy). A randomized mechanism $\mathcal{M}$ provides $(\epsilon, \delta)$-differential privacy (DP) if for any two neighboring datasets $D$ and $D'$ that differ in a single entry, $\forall S \subseteq Range(\mathcal{M})$,*

$$\Pr(\mathcal{M}(D) \in S) \leq e^\epsilon \cdot \Pr(\mathcal{M}(D') \in S) + \delta.$$

*where $\epsilon$ is the privacy budget and $\delta$ is the failure probability.*

Our added noise to $x_g$ is proportional to the sensitivity, as defined in Definition 3.3. The concept of sensitivity is originally used for sharing a dataset for achieving $(\epsilon, \delta)$-differential privacy. Later, we follow Theorem 3.4 to analyze the privacy on globally shared data.

**Definition 3.3.** *(Sensitivity). The sensitivity of a query function $\mathcal{F} : \mathbb{D} \to \mathbb{R}$ for any two neighboring datasets $D, D'$ is,*

$$\Delta = \max_{D, D'} \|\mathcal{F}(D) - \mathcal{F}(D')\|.$$

*where $\| \cdot \|$ denotes $L_1$ or $L_2$ norm.*

---

**Algorithm 1** FedAvg with **FedPD**

---

**server input:** initial $\phi^0$, communication round $R$
**client $k$'s input:** local epochs $E$, local datasets $\mathcal{D}^k$, learning rate $\eta_k$
  **Initialization:** server distributes the initial model $\phi^0$ to all clients,
  Globally shared dataset $\mathcal{D}^s$ generating. $\leftarrow$ Detail in Algorithm 2
  **Server Executes:**
  **for** each round $r = 1, 2, \cdots, R$ **do**
    server samples a subset of clients $\mathcal{S}_r \subseteq \{1, ..., K\}$, $n \leftarrow \sum_{i \in \mathcal{S}_r} |\mathcal{D}^i|$
    client $k$ samples a subet of globally shared dataset $\mathcal{D}_r^k \subseteq \mathcal{D}^s$ ($|\mathcal{D}_r^k| = |\mathcal{D}^k|$)
    server **communicates** $\phi^r$ to selected clients $k \in \mathcal{S}_r$ and sampled sharing data $\mathcal{D}_r^k$
    **for** each client $k \in \mathcal{S}_r$ **in parallel do**
      $\phi_{k,E-1}^{r+1} \leftarrow$ ClientTraining($k, \phi^r, \mathcal{D}_r^k$)
    **end for**
    $\phi^{r+1} \leftarrow \sum_{k \in \mathcal{S}_r}^{|\mathcal{S}_r|} \frac{|\mathcal{D}^i|}{n} \phi_{k,E-1}^r$
  **end for**

  **ClientTraining**($k, \phi, \mathcal{D}_r^k$)**:**
  **for** each local epoch $j$ with $j = 0, \cdots, E-1$ **do**
    $\phi_{k,j+1} \leftarrow \phi_{k,j} - \eta_k \nabla_\phi \mathcal{L}_F(\phi)$, i.e., Eq. 3
  **end for**
  **Return** $\phi$ to server

---

**Theorem 3.4.** *For any $\epsilon > 0, \delta \in [0, 1]$, and $\hat{\delta} \in [0, 1]$, the class of $(\epsilon, \delta)$-differentially private mechanisms satisfies $(\hat{\epsilon}_{\hat{\delta}}, 1 - (1 - \hat{\delta})\Pi_i(1 - \delta_i))$-differential privacy under $k$-fold adaptive composition for*

$$\hat{\epsilon}_{\hat{\delta}} = \min\{k\epsilon, (e^\epsilon - 1)\epsilon k/(e^\epsilon + 1) + \epsilon\sqrt{2k\log(e + \sqrt{k\epsilon^2/\hat{\delta}})}, (e^\epsilon - 1)\epsilon k/(e^\epsilon + 1) + \epsilon\sqrt{2k\log(1/\hat{\delta})}\}.$$

Since $\boldsymbol{x}_s$ is kept by the corresponding client, an adversary views nothing, which can be regarded as adding a sufficiently large noise on $\boldsymbol{x}$ to make it random enough. Considering all clients' data as a whole, we use a relatively small $\sigma$ (i.e., $\sigma_c < \sigma_d + \sigma_c$) for achieving much smaller privacy loss, summarized in Theorem 3.5.

**Theorem 3.5.** *Given identical privacy requirement, $\sigma_c$ of FedDP is much less than $\sigma$ that is supposedly added to raw data in conventional FL.*

Given $(\epsilon, \delta)$-DP at each client side, we utilize composition theorem to analyze overall privacy in FedPD. In summary, FedPD protects two types of data features using two different protective manners, i.e., small noise for generalizable features and extremely large noise for sensitive features, and thus attains higher model performance and stronger security in the same time.

## 4 EXPERIMENTS AND EVALUATION

### 4.1 EXPERIMENT SETUP

**Federated Non-IID Datasets.** We conduct experiments over various popular image classification datasets, including CIFAR-10, CIFAR100 (Krizhevsky et al., 2009), Fashion-MNIST(FMNIST) (Xiao et al., 2017), and SVHN (Netzer et al., 2011). We use latent dirichlet sampling (LDA) (Hsu et al., 2019) to simulate Non-IID distribution with 10 and 100 clients. The primary thought is to draw a $\mathbf{q} \sim Dir(\alpha\mathbf{p})$ from Dirichlet distribution, where $\alpha$ controls the heterogeneity degree. Here, the less $\alpha$ is, the more severe Non-IID distribution generate. In our experiments, we partition our datasets with two different degrees by LDA including $\alpha = 0.1$ and $\alpha = 0.05$. Besides, in order to prove that our framework works well under with Non-IID partitions. We also test other two kinds of partition strategy: (1) $\#C = k$ (McMahan et al., 2017; Li et al., 2022): each client only has $k$ different labels from dataset, and $k$ controls the unbalanced degree. (2) Subset method (Zhao et al., 2018): each client

Table 1: Results with/without FedPD on CIFAR-10

| | centralized training ACC = 95.48% w/(w/o) **FedPD** | | | | | | | |
|---|---|---|---|---|---|---|---|---|
| | ACC↑ | Gain↑ | Round↓ | Speedup↑ | ACC↑ | Gain↑ | Round↓ | Speedup↑ |
| | $\alpha = 0.1, E = 1, K = 10$ (Target ACC =79%) | | | | $\alpha = 0.05, E = 1, K = 10$ (Target ACC =69%) | | | |
| FedAvg | **92.34**(79.35) | 12.99↑ | **39**(284) | ×**7.3**(×1.0) | **90.02**(69.36) | 20.66↑ | **44**(405) | ×**9.2**(×1.0) |
| FedProx | **92.12**(83.06) | 9.06↑ | **62**(192) | ×**4.6**(×1.5) | **90.73**(78.98) | 11.75↑ | **48**(203) | ×**8.4**(×2.0) |
| SCAFFOLD | **89.66**(83.67) | 5.99↑ | **34**(288) | ×**8.4**(×1.0) | **81.04**(37.87) | 43.17↑ | **37**(None) | ×**10.9**(None) |
| FedNova | **92.23**(80.95) | 11.28↑ | **33**(349) | ×**8.6**(×0.8) | **91.21**(65.08) | 26.13↑ | **32**(None) | ×**12.7**(None) |
| | $\alpha = 0.1, E = 5, K = 10$ (Target ACC =85%) | | | | $\alpha = 0.1, E = 1, K = 100$ (Target ACC =49%) | | | |
| FedAvg | **93.24**(83.79) | 9.45 ↑ | **17**(261) | ×**15.4**(×1.0) | **84.06**(49.72) | 34.34↑ | **163**(967) | ×**5.9**(×1.0) |
| FedProx | **91.39**(82.32) | 8.97 ↑ | **76**(None) | ×**3.4**(None) | **87.01**(50.01) | 37.00↑ | **127**(831) | ×**7.6**(×1.2) |
| SCAFFOLD | **92.34**(85.31) | 7.03 ↑ | **15**(66) | ×**17.0**(×4.0) | **79.60**(52.76) | 26.84↑ | **171**(627) | ×**5.7**(×1.5) |
| FedNova | **92.85**(86.21) | 6.64 ↑ | **31**(120) | ×**8.4**(×2.2) | **86.64**(45.97) | 40.67↑ | **199**(None) | ×**4.9**(None) |

"Round" means the communication rounds that arrive at the target accuracy. ↓ and ↑ indicates smaller (larger) values are better. "None" implies not attaining the target accuracy during the entire training process. All the "Speedup" is calculated by comparing with vanilla FedAvg "Round" in different Non-IID partition scenarios.

has all classes from the data, but one dominant class far away outnumbers other classes. These three partition methods mainly include label skew and quantity skew. The visualization of data distribution is shown in Figure 4 in Appendix A.1.

**Models, Metrics and Baselines.** We use ResNet-18 (He et al., 2016) both in content extractor during sharing data generation process and classifier in FL. And we exploit $\beta$-VAE (Higgins et al., 2016) for privacy distiller for privacy-preserving. We evaluate the model performance on two popular metrics in FL, i.e. the communication rounds to reach the target accuracy and the best accuracy in whole training process. Note that the target accuracy is set as the best accuracy of vanilla FedAvg in different scenarios. We conduct FedAvg (McMahan et al., 2017) and some other popular methods including FedProx (Li et al., 2020), SCAFFOLD (Karimireddy et al., 2020), FedNova (Wang et al., 2020b), with or without FedPD, to explore the potency of our method. We conduct all algorithms with local epochs $E = 1$ and $E = 5$. The detailed hyper-parameters of each FL algorithm and privacy distillation in different datasets are listed in A.3.1.

## 4.2 EXPERIMENTAL RESULTS

**Main Results.** The results on CIFAR-10, CIFAR-100, FMNIST, and SVHN are shown respectively in Tables 1, 2, 5, and 6, which demonstrates that FedPD has a significant performance gain. We also show the convergence speed of different algorithms on CIFAR-10 with $a = 0.1$, $E = 1$, $M = 10$ in Figure 2a,[1] which shows that FedPD can also greatly improve the convergence rate.

Table 2: Results with/without FedPD on CIFAR-100

| | centralized training ACC = 75.56% w/(w/o) **FedPD** | | | | | | | |
|---|---|---|---|---|---|---|---|---|
| | ACC↑ | Gain↑ | Round↓ | Speedup↑ | ACC↑ | Gain↑ | Round↓ | Speedup↑ |
| | $\alpha = 0.1, E = 1, K = 10$ (Target ACC =67%) | | | | $\alpha = 0.05, E = 1, K = 10$ (Target ACC =61%) | | | |
| FedAvg | **69.64**(67.84) | 1.8↑ | **283**(495) | ×**1.7** (×1.0) | **68.49**(62.01) | 6.48↑ | **137**(503) | ×**3.7**(×1.0) |
| FedProx | **70.02**(65.34) | 4.68 ↑ | **233**(None) | ×**2.1**(None) | **69.03**(61.29) | 7.74↑ | **141**(485) | ×**3.6**(1.0) |
| SCAFFOLD | **70.14**(67.23) | 2.91↑ | **198**(769) | ×**2.5**(× 0.6) | **69.32**(58.78) | 10.54↑ | **81**(None) | ×**6.2**(None) |
| FedNova | **70.48**(67.98) | 2.5↑ | **147**(432) | ×**3.4**(×1.1) | **68.92**(60.53) | 8.39↑ | **87**(None) | ×**5.8**(None) |
| | $\alpha = 0.1, E = 5, K = 10$ (Target ACC =69%) | | | | $\alpha = 0.1, E = 1, K = 100$ (Target ACC =48%) | | | |
| FedAvg | **70.96**(69.34) | 1.62↑ | **79**(276) | ×**3.5**(×1.0) | **60.58**(48.21) | 12.37↑ | **448**(967) | ×**2.2**(×1.0) |
| FedProx | **69.66**(62.32) | 7.34↑ | **285**(None) | ×**1.0**(None) | **67.69**(48.78) | 18.91↑ | **200**(932) | ×**4.8**(×1.0) |
| SCAFFOLD | **70.76**(70.23) | 0.53↑ | **108**(174) | ×**2.6**(×1.6) | **66.67**(51.03) | 15.64↑ | **181**(832) | ×**5.3**(×1.2) |
| FedNova | **69.98**(69.78) | 0.2↑ | **89**(290) | ×**3.1**(×1.0) | **67.62**(48.03) | 19.59↑ | **198**(976) | ×**4.9**(×1.0) |

---

[1]More figures of convergence speed of other experiments are shown in appendix A.4.

Table 3: Experiment results of different Non-IID partition methods on CIFAR-10 with 10 clients.

| Partition Method | Test Accuracy w/(w/o) **FedPD** | | | |
|---|---|---|---|---|
| | FedAvg | FedProx | SCAFFOLD | FedNova |
| $\alpha = 0.1$ | **92.34**(79.35) | **92.12**(83.06) | **89.66**(83.67) | **92.23**(80.95) |
| $\#C = 2$ | **89.23**/42.54 | **88.17**/58.45 | **84.43**/46.82 | **89.54**/45.42 |
| Subset | **90.29**/39.53 | **89.11**/32.87 | **89.92**/35.26 | **90.00**/38.52 |

Table 4: Experiment results with different noise adding in CIFAR-10.

| Noise Type | Test Accuracy on Different Noise | | | |
|---|---|---|---|---|
| | FedAvg | FedProx | SCAFFOLD | FedNova |
| Gaussian Noise | 92.34 | 92.12 | 89.66 | 92.23 |
| Laplacian Noise | 92.30 | 91.36 | 91.24 | 91.73 |

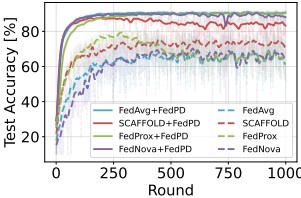

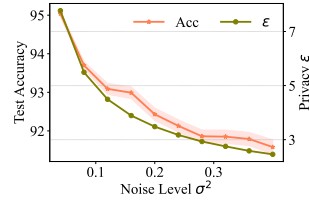

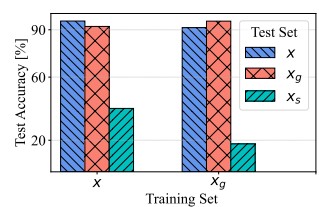

(a) Test Accuracy on CIFAR-10 with $\alpha = 0.1, E = 1, K = 10$ and Gaussian Filter for better visualization.

(b) Test Accuracy on FMNIST with different noise level $\sigma^2$, obtaining various privacy $\epsilon$(lower $\epsilon$ is preferred).

(c) Two clasifiers trained on different data form and test accuracy on $x$, $x_s$, and $x_g$ respectively.

Figure 2: Experiments of the relationship between privacy and performance.

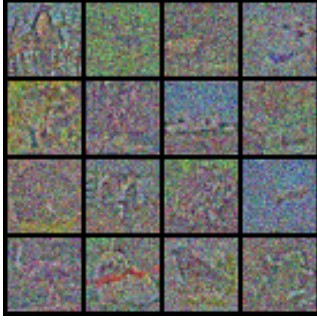

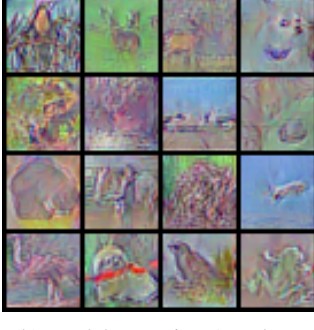

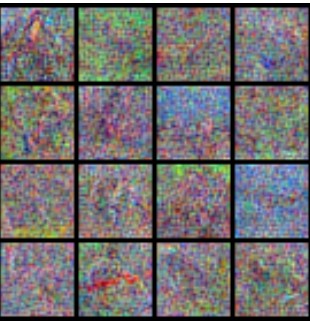

(a) Globally Shared Data $\boldsymbol{x}_p$

(b) Model Inversion Attack $\boldsymbol{x}_g$

(c) Model Inversion Attack $\boldsymbol{x}_p$

Figure 3: Model Inversion Attack Results. White-Box attack globally shared data $\boldsymbol{x}_p$ and generalizable features $\boldsymbol{x}_g$, respectively. The result of being attacked is in (b) and (c) to compare with shared data $\boldsymbol{x}_p$ in (a)

**Privacy and performance.** To explore the relationship between privacy level $\epsilon$ and performance, we conduct experiments with different $\sigma^2$. As shown in Figure 2b, the performance decreases with the increasing protection strength. Another Laplacian noise report comparable results with Gaussian noise listed in Table 4. In conclusion, we suggest sacrificing part of the privacy when encountering limited communication resources. Another question is, can the globally shared data be inferred by some attack methods? To answer this question, we resort model inversion attack (He et al., 2019), widely used in the literature to reconstruction our shared data. The results on Figure 3b indicates that only privacy distillation still have risk of privacy leakage. Figure 3c also be a strong testimony for the differential privacy of noise adding on generalizable features. Furthermore, FedPD can give a strong private information protection. The original image can be found in Appendix A.2

**Different number of clients.** Table 1, Table 2, Table 5, and Table 6 show that FedPD strengthen the performance and speed up the convergence both in 10 and 100 clients. Especially 100 clients in CIFAR-10 and CIFAR100 have a noteworthy enhancement. The reason may be that FL on CIFAR-10 and CIFAR100 with 100-clients has more diverge data distribution than FMNIST. With FedPD, the missed data knowledge can be well replenished.

**Different data heterogeneity.** Table 1, Table 2, Table 5, and Table 6 show that high Non-IID degree ($\alpha$=0.05) achieve a better improvement than lower unbalanced degree ($\alpha$=0.1), which also indicates that FedPD can well defend against data heterogeneity. Moreover, Table 3 shows that other two kinds of heterogeneity partition cause more performance decline compared with LDA ($\alpha = 0.1$), and FedPD attains comparable improvement with LDA $\alpha = 0.1$, indicating FedPD is insensitive to other Non-IID data distribution.

Table 5: Results with/without FedPD on FMNIST

| | centralized training ACC = 95.64% w/(w/o) **FedPD** | | | | | | | |
|---|---|---|---|---|---|---|---|---|
| | ACC↑ | Gain↑ | Round↓ | Speedup↑ | ACC↑ | Gain↑ | Round↓ | Speedup↑ |
| | $\alpha = 0.1, E = 1, K = 10$ (Target ACC =86%) | | | | $\alpha = 0.05, E = 1, K = 10$ (Target ACC =78%) | | | |
| FedAvg | **92.34**(86.73) | 5.61↑ | **14**(121) | ×**8.6**(×1.0) | **90.69**(78.34) | 12.35↑ | **16**(420) | ×**26.3**(×1.0) |
| FedProx | **92.09**(87.73) | 4.36↑ | **32**(129) | ×**2.1**(×0.9) | **89.68**(82.03) | 7.65↑ | **16**(44) | ×**26.3**(9.5) |
| SCAFFOLD | **91.62**(86.31) | 3.89↑ | **29**(147) | ×**4.2**(×0.8) | **80.48**(76.63) | 3.85↑ | **139**(None) | ×**6.2**(None) |
| FedNova | **92.39**(87.03) | 5.36↑ | **18**(88) | ×**6.7**(×1.4) | **89.72**(79.98) | 9.74↑ | **16**(531) | ×**26.3**(×0.8) |
| | $\alpha = 0.1, E = 5, K = 10$ (Target ACC =87%) | | | | $\alpha = 0.1, E = 1, K = 100$ (Target ACC =90%) | | | |
| FedAvg | **92.26**(87.43) | 4.83↑ | **19**(276) | ×**14.5**(×1.0) | **92.71**(90.21) | 2.5↑ | **243**(687) | ×**2.8**(×1.0) |
| FedProx | **91.79**(86.63) | 5.16↑ | **34**(None) | ×**8.1**(None) | **92.82**(90.17) | 2.65↑ | **284**(501) | ×**2.4**(×1.4) |
| SCAFFOLD | **92.92**(87.21) | 5.71↑ | **8**(112) | ×**34.5**(×2.5) | **90.28**(84.87) | 5.41↑ | **952**(None) | ×**0.7**(None) |
| FedNova | **92.30**(87.67) | 4.63↑ | **8**(187) | ×**34.5**(×1.5) | **91.04**(85.32) | 5.72↑ | **589**(None) | ×**1.2**(None) |

Table 6: Results with/without FedPD on SVHN

| | centralized training ACC = 96.56% w/(w/o) **FedPD** | | | | | | | |
|---|---|---|---|---|---|---|---|---|
| | ACC↑ | Gain↑ | Round↓ | Speedup↑ | ACC↑ | Gain↑ | Round↓ | Speedup↑ |
| | $\alpha = 0.1, E = 1, K = 10$ (Target ACC =88%) | | | | $\alpha = 0.05, E = 1, K = 10$ (Target ACC =82%) | | | |
| FedAvg | **93.21**(88.34) | 4.87↑ | **105**(264) | ×**2.5**(×1.0) | **93.49**(82.76) | 10.73↑ | **194**(365) | ×**1.9**(×1.0) |
| FedProx | **91.80**(86.23) | 5.574↑ | **233**(None) | ×**1.1**(None) | **93.21**(79.43) | 13.78↑ | **37**(None) | ×**9.9**(None) |
| SCAFFOLD | **88.41**(80.12) | 8.29↑ | **357**(None) | ×**0.**(None) | **90.27**(75.87) | 14.4↑ | **64**(None) | ×**5.7**(None) |
| FedNova | **92.98**(89.23) | 3.75↑ | **113**(276) | ×**2.3**(×1.0) | **93.05**(82.32) | 10.73↑ | **37**(731) | ×**9.9**(×0.5) |
| | $\alpha = 0.1, E = 5, K = 10$ (Target ACC =87%) | | | | $\alpha = 0.1, E = 1, K = 100$ (Target ACC =89%) | | | |
| FedAvg | **93.77**(87.24) | 6.53↑ | **105**(128) | ×**1.2**(×1.0) | **91.04**(89.32) | 1.72↑ | **763**(623) | ×0.8(×**1.0**) |
| FedProx | **91.15**(77.21) | 13.94↑ | **142**(None) | ×**0.9**(None) | **91.41**(88.76) | 2.65↑ | **733**(645) | ×0.8(×**1.0**) |
| SCAFFOLD | **93.78**(80.98) | 12.8↑ | **20**(None) | ×**6.4**(None) | **92.73**(88.32) | 4.41↑ | **507**(687) | ×**1.2**(×0.9) |
| FedNova | **93.66**(89.03) | 4.63↑ | **52**(177) | ×**2.5**(×0.7) | **84.05**(81.87) | 2.18↑ | None(None) | None(None) |

**Different local epochs.** To test the effect of local epoch $E$, we choose $E = 1$ and $E = 5$ with the same Non-IID degree ($\alpha = 0.1$) and client number ($K = 10$). We run 1000 rounds with 1 epoch local training and 400 rounds for 5 epochs local update. The results show that FedPD is robust to the local epochs.

**Other Facts of FedPD.** For a intuitive understanding of why we utilize $x_g$ as a substitute of raw data $x$ without drastic performance degradation. We train two different networks separately on $x$ and $x_g$ on CIFAR-10 and test them on $x$, $x_s$, and $x_g$, respectively. The results presented in Figure 2c. As we can see, the most useful features for downstream tasks are contained in $x_g$. More experimental details are presented in Appendix A.4

## 5 CONCLUDING REMARKS

In this paper, we observe that model gains a substantial performance assisted by generalizable features. Later we conduct DP to protect generalizable features and contruct a globally shared dataset for defying heterogeneity in FL. Our contribution lies in not only improving model performance in Non-IID scenarios, but also inspiring a new viewpoint on data-dominated secure sharing, e.g., distillation data before knowledge learning. We expect that our work could simulated further data-dominated sharing in FL or other popular learning algorithms.

Our framework shows suprior results against model inversion attack, yet we have not finished exploring data poisoning attack, given the shared data. We conduct preliminary experiments on data poisoning attacks, in which some clients send Gaussian noise to the server, causing performance degradation and slow convergence. Limited storage or communication resources may limited the power of FedPD, since FedPD introduces extra storage overhead. We leave it as our future work to explore the storage-friendly FedPD.

## ETHIC STATEMENT

This paper does not raise any ethical concerns. This study does not involve any human subjects, practices to data set releases, potentially harmful insights, methodologies and applications, potential conflicts of interest and sponsorship, discrimination/bias/fairness concerns, privacy and security issues, legal compliance, and research integrity issues.

## REPRODUCIBILITY STATEMENT

To make all experiments reproducible, we have listed all detailed hyper-parameters of each FL algorithm and privacy distillation on different datasets in A.3.1. Due to the privacy concerns, we will upload the anonymous link of source codes and instructions during the discussion phase to make it only visible to reviewers. All definitions can be found in Section 3. And the complete proof can be found in Appendix A.6.

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

## A  APPENDIX

### A.1  VISUAL

We show the visualization of data distribution in Figure 4. The LDA partition and the $\#C = 2$ partition have the label skew and the quantity skew simultaneously. And the Subset partition only has the label skew.

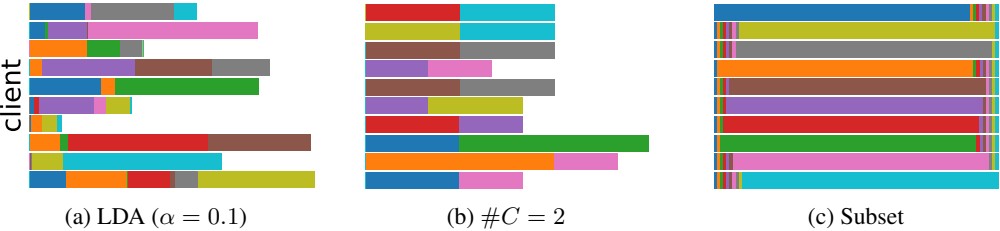

(a) LDA ($\alpha = 0.1$)        (b) $\#C = 2$        (c) Subset

Figure 4: Data distribution in various FL heterogeneity scenario. Different colors denote different labels and the length of each line denote data number.

### A.2  GLOBALLY SHARED DATA

We display the globally shared data $x_p$ from four different datasets and the raw data $x$ to compare our privacy protection. Firstly, the raw data in Figure 3 shown in Figure 5.

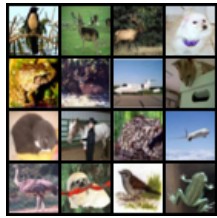

Figure 5: Raw Data in Model Inversion Attack.

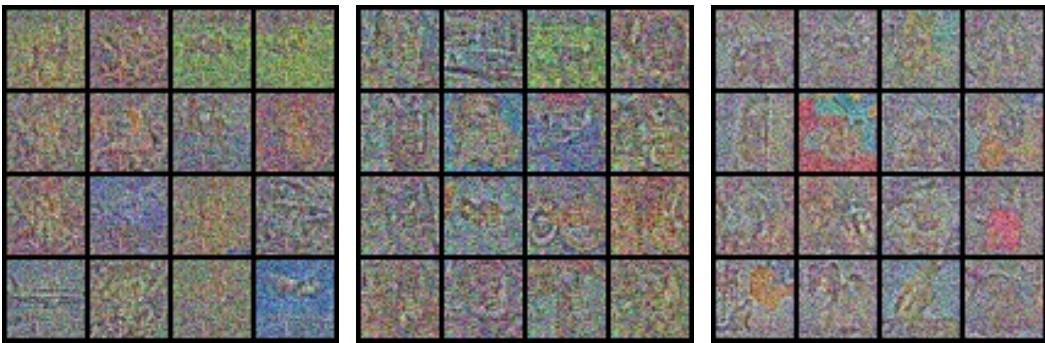

Figure 6: Globally Shared Data $x_p$ on CIFAR-10.

### A.3  MORE DETAILS OF FEDPD

Algorithm 1 give us an intuitive explanation of how we deploy FedPD on FL algorithm e.g., FedAvg and Algorithm 2 illustrates the procedure to generate globally shared data.

---

**Algorithm 2** Globally Shared Data Generation

---

**Server input:** generation process communication round $T$, noise mean $\mu$, noise level $\sigma^2$
**Client $k$'s input:** local epochs $Q$, local datasets $\mathcal{D}^k$

  **Initialization:** server distributes the initial model $w^0, \theta^0$ to all clients,
  **Server Executes:**
  **for** each round $t = 1, 2, \cdots, T$ **do**
    server samples a subset of clients $\mathcal{S}_g \subseteq \{1, ..., K\}$, $n \leftarrow \sum_{i \in \mathcal{S}_g} |\mathcal{D}^i|$
    server **communicates** $w^t, \theta^t$ to all clients
    **for** each client $k \in \mathcal{S}_g$ **in parallel do**
      $w^{t+1}_{k,q-1} \theta^{t+1}_{k,q-1} \leftarrow$ ClientGenerationTraining$(k, w^t, \theta^t, \mu, \sigma^2)$
    **end for**
    $w^{t+1}, \theta^{t+1} \leftarrow \sum_{k \in \mathcal{S}_g}^{|\mathcal{S}_g|} \frac{|\mathcal{D}^i|}{n} w^t_{k,Q-1}, \sum_{k \in \mathcal{S}_g}^{|\mathcal{S}_g|} \frac{|\mathcal{D}^i|}{n} \theta^t_{k,Q-1}$
  **end for**
  **for** all clients $k$ with $k = 0, \cdots, K$ **do**
    $\mathcal{D}^s_k \leftarrow$ SharedDataGeneration$(w_k, \theta_k, \mu, \sigma^2)$
    send $\mathcal{D}^s_k$ to server to construct globally shared dataset $\mathcal{D}^s$
  **end for**

  **ClientGenerationTraining**$(k, w, \theta, \mu, \sigma^2)$**:**
  **for** each local epoch $q$ with $q = 0, \cdots, Q-1$ **do**
    $w_{k,q+1}, \theta_{k,q+1} \leftarrow$ PrivacyDistillation$(w_k, \theta_k, \mu, \sigma^2)$ using Eq.1
  **end for**
  **Return** $w_{k,Q-1}, \theta_{k,Q-1}$ to server

---

### A.3.1 HYPER-PARAMETERS

We fine-tuned learning rates in $0.0001, 0.001, 0.01, 0.1$ and report the best results and corresponding learning rate. In most case, we use $0.01$ as the learning rate except SCAFFOLD and FedNova in SVHN under the $\alpha = 0.1, E = 1, K = 100$ setting, the learning rate is $0.0001$ and $0.001$, respectively. Batch size is set as $64$ in when $K = 10$ and $32$ for $K = 100$. The number of clients selected for aggregation on server side is $5$ per round for $K = 10$, and $10$ for $K = 100$. The noise level in our experients is $\mathcal{N}(0, 0.15)$

### A.3.2 TRICK FOR FEDPD

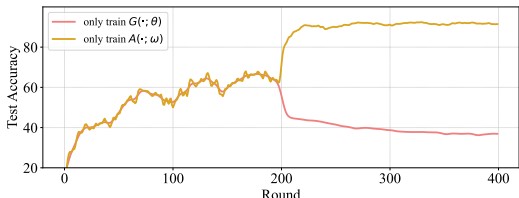

Figure 7: The pink line indicates $6.25\%$ dataset to train VAE and Auxiliary 200 rounds jointly and the whole dataset for only VAE $G(\cdot; \theta)$ training 200 rounds in the following. Furthermore, the yellow one uses the same data in the former 200 rounds as the pink line but the complete dataset to train Auxiliary Classifier $A(\cdot; \omega)$.

In addition, we provide an insight experiment on the need for mixupdata (Zhang et al., 2017) augmentation in our approach shown in Figure 7. As we can see, the absence of data leads to poor generalization of the auxiliary classifier $A$ on $x$ and adequate data for VAE $G$ still has a bad effects.

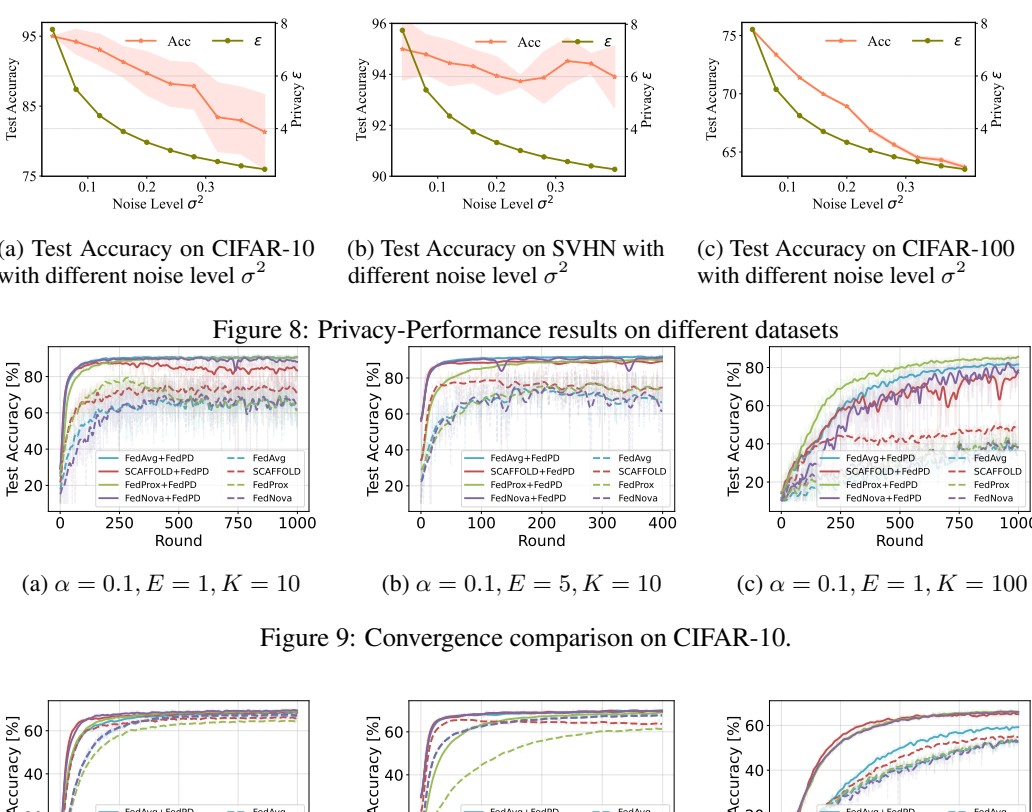

(a) Test Accuracy on CIFAR-10 with different noise level $\sigma^2$

(b) Test Accuracy on SVHN with different noise level $\sigma^2$

(c) Test Accuracy on CIFAR-100 with different noise level $\sigma^2$

Figure 8: Privacy-Performance results on different datasets

(a) $\alpha = 0.1, E = 1, K = 10$

(b) $\alpha = 0.1, E = 5, K = 10$

(c) $\alpha = 0.1, E = 1, K = 100$

Figure 9: Convergence comparison on CIFAR-10.

(a) $\alpha = 0.1, E = 1, K = 10$

(b) $\alpha = 0.1, E = 5, K = 10$

(c) $\alpha = 0.1, E = 1, K = 100$

Figure 10: Convergence comparison on CIFAR-100.

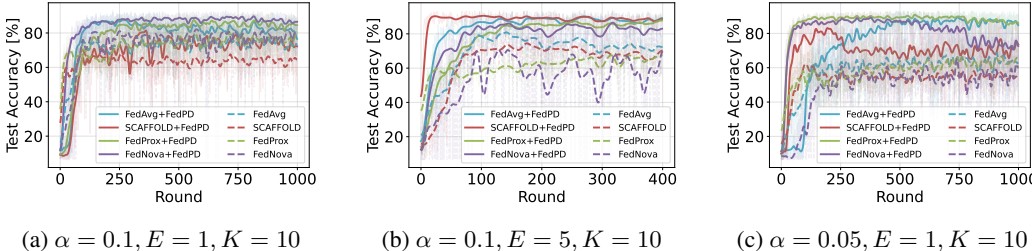

(a) $\alpha = 0.1, E = 1, K = 10$

(b) $\alpha = 0.1, E = 5, K = 10$

(c) $\alpha = 0.05, E = 1, K = 10$

Figure 11: Convergence comparison on SVHN.

## A.4 MORE EXPERIMENT RESULTS

## A.5 MORE RELATED WORK

**Federated Learning with heterogeneous data.** In FL, all distributed clients jointly train a model across various distributed datasets for user privacy protection, while local data is not accessible to other clients. FedAvg (McMahan et al., 2017) is the first work proposed to reduce communication overhead and preserve privacy by more local training epochs and fewer communication rounds. However, some studies (Zhao et al., 2018; Li et al., 2022) have pointed out that the divergence between FedAvg and centralized training is slight in the IID case. But, in heterogeneous distribution, there is a considerable divergence between the different clients and centralized training, and the gap

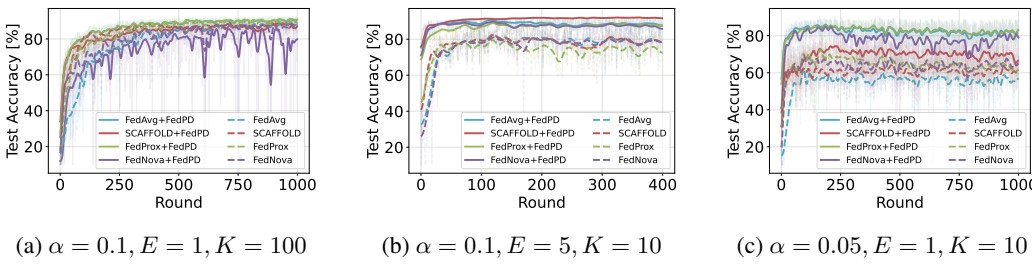

Figure 12: Convergence comparison on FMNIST.

accumulates during the FedAvg weighted aggregation, leading to the performance degradation of FL models.

Recently, a series work propose new learning objective to calibrate the update direction of local training from being too far away from the global model. FedProx (Li et al., 2020) adds a $L_2$ distance as the regularization term in the objective function and provides a theoretical guarantee of convergence. Similarly, a novel objective function is also introduced in FedIR (Hsu et al., 2020) over a mini-batch by self-normalized weights to address the non-identical class distribution. SCAFFOLD (Karimireddy et al., 2020) restricts the model using previous information. Besides, MOON (Li et al., 2021a) introduces constrastive learning at the model level to correct the divergence between clients and server.

Meanwhile, recent works propose designing new model aggregation schemes. FedAvgM (Hsu et al., 2019) performs momentum on the server side. FedNova (Wang et al., 2020b) adopts normalized averaging method to eliminate objective inconsistency. A study (Cho et al., 2020) also indicates that biasing client selection with higher local loss can speed up the convergence rate. The coordinate-wise averaging of weights also induce noxious performance. FedMA (Wang et al., 2020a) conducts Bayesian non-parametric strategy for heterogeneous data. FedBN (Li et al., 2021c) focus on feature shift Non-IID and perform local batch normalization before averaging models.

Another existing direction for tackling data heterogeneity is sharing data. This line of works mainly to assemble the data of different clients to construct a global IID dataset, mitigating client drift by replenishing the lack of information of clients (Zhao et al., 2018). Existing methods include synthesizing data based on the raw data by GAN (Jeong et al., 2018; Long et al., 2021). However, the synthetic data is generally relatively similar to the raw data, leading to privacy leakage at some degree. Adding a noise to the shared data is another promising strategy (Chatalic et al., 2022; Cai et al., 2021). Some methods employ the statistics of data (Yoon et al., 2021; Shin et al., 2020) to synthesize for sharing, which still contains some raw data content. Other methods distribute intermediate features (Hao et al., 2021), logits (Chang et al., 2019; Luo et al., 2021), or learn the new embedding (Tan et al., 2022). These tactics will increase the difficulty of privacy protection because some existing methods can reconstruct images based on feature inversion methods (Zhao et al., 2020). Most of the above methods share information without a privacy guarantee or with strong privacy-preserving but poor performance, posing the privacy-performance dilemma.

Concretely, in FD (Jeong et al., 2018) all clients leverages a generative model collaboratively for data generation in a homogeneous distribution. For a better privacy protection, G-PATE (Long et al., 2021) performs discriminators with local aggregation in GAN. Fed-ZDAC(Fed-ZDAS) (Hao et al., 2021), depending on which side to play augmentation, introduce zero-shot data augmentation by gathering intermediate activations and batch normalization(BN) statistics to generate fake data. Inspired by mixup data, MAFL (Yoon et al., 2021) propose FedMIx to share information by averaging local data which also brings about the privacy problem. Cronus (Chang et al., 2019) transmit the logits information while CCVR (Luo et al., 2021) collect statistical inforamtion of logits to sample fake data. FedFTG (Zhang et al., 2022) use a generator to explore input space of local model and transfer local knowledge to global model. FedDF (Lin et al., 2020) utilizes knowledge distillation based on unlabeled data or a generator and then conduct *AVGLOGITS*. The main difference between FedDF and FedPD is that our method distill the privacy kept locally rather than distilling knowledge. We provide multi steps to protect privacy with drastic performance gain.

**Differential privacy with federated learning**. Recent works on model memorization and gradient leakage confirms that model parameters are seemingly secure. Carlini et.al (Carlini et al., 2019) found that unintended-and-persistent memorization of sensitive data occurs early during training with no relation to data rarity and model size. Training with differential privacy (Zhu et al., 2019)(Nasr et al., 2019) is a feasible solution to avoid serious consequences, albeit at some loss in utility.

Differential privacy is a framework to quantify to what extent individual privacy in a statistical dataset is preserved while releasing the established model over specific datasets. It has spawned a large set of research topics in data-releasing mechanism and noise-adding mechanism. Particularly, noise-adding mechanism has been widely utilized in various differentially private learning algorithms for protecting whether an individual is in the dataset or not.

In federated settings, training with differential privacy, i.e., adding noise to the model/data, originally aims to protect local information of each clients. Say, an adversary should not be able to discern whether a client's data was used for early training. Here, we summarize some works with high citation or from top venue. Yuan et al (Yuan et al., 2019) apply differential privacy to protect medical images by adopting famous AlexNet and Gaussian mechanism. Huang et al (Huang et al., 2020) integrate an approximate augmented Lagrangian function and Gaussian noise mechanism for balancing utility and privacy in FL. Wei etal (Wei et al., 2020) perturb early-trained parameters locally by adding noises before uploading them to a server for aggregation. Both Huang et al and Wei et al are first (to their knowledge) to analyze the relation between convergence and utility in FL. Andrew et al (Thakkar et al., 2019) explore to set an adaptive clipping norm in federated setting rather than using a fixed one. They show that adaptive clipping to gradients can perform as well as any fixed clip chosen by hand.

Kim et al (Kim et al., 2021) provide a noise variance bound that guarantees local DP after multiple rounds of parameter aggregations. They introduce a trilemma in privacy, utility, and transmission rate of a federated stochastic gradient decent. Hoeven et al (van der Hoeven, 2019) introduce data-dependent bounds and apply symmetric noise in online learning, which allows data provider to pick noise distribution. Triastcyn et al (Triastcyn & Faltings, 2019) adapt the notion of Bayesian differential privacy to federated learning and make necessary analyses on privacy guarantee. Sun et al (Sun et al., 2021) explicitly vary ranges of weights at different layers in a DNN, and shuffle high-dimensional parameters at an aggregation for easing explodes of privacy budgets. All works above start to apply DP and its variants to federated setting for different goals/scenarios, which thus provide underlying security as *DP* guarantees.

A.6    DIFFERENTIAL PRIVACY

Proof of Theorem 3.4 is here.

*Proof.*

**Definition A.1.** (*Privacy Loss*). *Let $\mathcal{M} : \mathbb{D} \to \mathbb{R}$ be a randomized mechanism with input domain $D$ and range $R$. Let $D, D'$ be a pair of adjacent dataset and* aux *be an auxiliary input. For an outcome $o \in \mathbb{R}$, the privacy loss at $o$ is defined by:*

$$\mathcal{L}_{\mathsf{Pri}}^{(o)} \triangleq \log \frac{\Pr[\mathcal{M}(\mathsf{aux}, D) = o]}{\Pr[\mathcal{M}(\mathsf{aux}, D') = o]} \tag{4}$$

We need to compute the privacy loss on an outcome $o$ as a random variable when the random mechanism operates on two adjacent database $D$ and $D'$. Privacy loss is a random variable that accumulates the random noise added to the algorithm/model.

We aim at an exact analysis on privacy via compositing multiple random mechanisms. For simplification, we start with a particular random mechanism $\mathcal{M}^{\dagger}$ and then generalize it. The mechanism $\mathcal{M}^{\dagger}$ does not depend on database or the query but relies on hypothesis hp. For hp $= 0$, the outcome $O_i$ of $\mathcal{M}_i^{\dagger}$ is independent and identically distributed from a discrete random distribution $O^{\mathsf{hp}=0} \sim \mathcal{P}^{\dagger,0}$. $\mathcal{P}^{\dagger,0}(o)$ is defined to be: $\delta$ for $o = 0$; $(1 - \delta)e^{\epsilon}/(1 + e^{\epsilon})$ for $o = 1$; $(1 - \delta)/(1 + e^{\epsilon})$ for $o = 2$; $0$ for $o = 3$. For hp $= 1$, the outcome $O_i$ of $\mathcal{M}_i^{\dagger}$ is $O^{\mathsf{hp}=1} \sim \mathcal{P}^{\dagger,1}$. $\mathcal{P}^{\dagger,1}(o)$ is defined to be: $0$ for $o = 0$; $(1 - \delta)/(1 + e^{\epsilon})$ for $o = 1$; $(1 - \delta)e^{\epsilon}/(1 + e^{\epsilon})$ for $o = 2$; $\delta$ for $o = 3$.

Let $\mathcal{R}(\epsilon, \delta)$ be privacy region of a single access to $\mathcal{M}^{\dagger}$. Privacy region consists of two rejection regions with errors, i.e., rejecting true null-hypothesis (type-I error) and retaining false null-hypothesis

(type-II error). Let $\epsilon_k^\dagger, \delta_k^\dagger$ be $\mathcal{M}_i^\dagger$'s parameters for defining privacy. $\mathcal{R}(\mathcal{M}, D, D')$ of any mechanism $\mathcal{M}$ can be regarded as an intersection of $\{(\epsilon_k^\dagger, \delta_k^\dagger)\}$ privacy regions. For an arbitrary mechanism $\mathcal{M}$, we need to compute its privacy region using the $(\epsilon_k^\dagger, \delta_k^\dagger)$ pairs. Let $D, D'$ be neighboring databases and $\mathcal{O}$ be the outputting domain. Define (symmetric) $\mathcal{P}, \mathcal{P}'$ to be probability density function of the outputs $\mathcal{M}(D), \mathcal{M}(D')$, respectively. Assume a permutation $\pi$ over $\mathcal{O}$ such that $\mathcal{P}'(o) = \mathcal{P}(\pi(o))$. Let $S$ denote the complement of a rejection region. Since $\mathcal{R}(\mathcal{M}, D, D')$ is convex, we have

$$1 - \mathcal{P}(S) \geq -e^{\epsilon_k^\dagger} \mathcal{P}'(S) + 1 - \delta_k^\dagger \Rightarrow \mathcal{P}(S) - e^{\epsilon_k^\dagger} \mathcal{P}'(S) \leq \delta_k^\dagger$$

Define $\mathsf{Dt}_{\epsilon^\dagger}(\mathcal{P}, \mathcal{P}') = \max_{S \subseteq \mathcal{O}} \{\mathcal{P}(S) - e^{\epsilon^\dagger} \mathcal{P}'(S)\}$. Thus, $\mathcal{M}$'s privacy region is the set: $\{(\epsilon_k^\dagger, \delta_k^\dagger) : \epsilon_k^\dagger \in [0, \infty)]$ s.t. $\mathcal{P}(o) = e^{\epsilon_k^\dagger} \mathcal{P}'(o), \delta_k^\dagger = \mathsf{Dt}_{\epsilon_k^\dagger}(\mathcal{P}, \mathcal{P}')\}$. Next, we consider composition on random mechanisms $\mathcal{M}_1, \ldots, \mathcal{M}_i$. By accessing $\mathcal{M}_i^\dagger$, $\mathcal{P}(O^{1,\mathsf{hp}} = o_1, \ldots, O^{i,\mathsf{hp}} = o_i) = \Pi_{j=1}^i \mathcal{P}^{\dagger,\mathsf{hp}}(o_j)$. By algebra on two discrete distributions,

$$\mathsf{Dt}_{(i-2j)\epsilon}(\mathcal{P}^i, (\mathcal{P}')^i) = 1 - (1-\delta)^i + (1-\delta)^i \sum_{l=0}^{j-1} \left( \binom{i}{l} \left( e^{\epsilon(i-l)} - e^{\epsilon(i-2j+l)} \right) \right) / (1 + e^\epsilon)^k$$

Hence, privacy region is an interaction of $i$ regions, parameterized by $1 - (1-\hat{\delta})\Pi_i(1-\delta_i)$. $\quad\square$

