# OpenReview forum: "FedPD: Defying data heterogeneity through privacy distillation"
_ICLR.cc/2023/Conference — Submitted to ICLR 2023_

### Official Review · Reviewer_52C4 · 2022-10-19

**Confidence:** 4
**Correctness:** 1
**Technical Novelty And Significance:** 1
**Empirical Novelty And Significance:** 1
**Recommendation:** 1

**Clarity, Quality, Novelty And Reproducibility:**

I think the paper scores poorly on all four of these aspects. Specific to reproducibility, the proofs and pseudocode are vague, and no experiment code is provided.

**Details Of Ethics Concerns:**

Theorem 3.4 in this paper appears to be plagiarized from Theorem 3.4 and Theorem 3.5 in the Kairouz+ paper about advanced composition, https://arxiv.org/pdf/1311.0776.pdf. Theorem 3.4 in this paper mixes the two together (homogeneous epsilon, heterogeneous delta) but the wording and notation is nearly identical. This paper, however, does not cite Kairouz+ or give any indication that the result is not original to this paper. The proof of Theorem 3.4 given in this paper is also vague and imprecise enough to be, IMO, "not even wrong".

EDIT: The authors have promised to add a citation to this result and clarify that it is not a contribution of this work, so I'm removing the Ethics Review flag. However, I'm going to leave the previous entry here to reflect how the original submission changed.

**Strength And Weaknesses:**

Strengths: The idea of separating useful and sensitive features is conceptually interesting.

Weaknesses: I think the paper has three serious problems.

1. The description of the algorithm is unclear. After reading Section 3.2 several times, I still don't understand how the data distillation process works. Is it performed entirely locally? If so, what is the meaning of the E_{(x,y)} term? As far as I can tell, the paper never makes the actions taken by each client locally explicit. Algorithm 1 abstracts it away as a subroutine in Algorithm 2 in the appendix, and Algorithm 2 abstracts it away as calling an undefined PrivacyDistillation algorithm using Equation 1, though I don't see where the user's actual data point enters into the algorithm. If indeed the distillation process is entirely local, I'm confused about how generalizability is even measured -- how can somebody with one data point reason about which parts of it generalize? Do users have multiple data points? Turning to the privatization step, I do not see which part of the paper actually derives the additive noise scale required to privatize the generalizable features x_g when constructing the globally shared database. The relevant result appears to be Theorem 3.5, which only offers a vague qualitative statement that FedPD's noise standard deviation is "much less than" that of conventional federated learning, without a proof. Overall, the paper's clarity is low.

2. The privacy argument seems wrong. As far as I understand it, the paper does not attempt to ensure that the distillation process itself is private. This makes me question the value of adding noise to the distilled generalizable features. The paper is not precise enough for me to make this argument exact, but it seems plausible that the scale of the generalizable feature is itself private information, i.e., certain data points will produce generalizable features with noticeably larger norms than other data points. If indeed noise is scaled to this norm (as I assume it must be -- if not, what is the advantage over privatizing the entire data point?) then the scale of the noise added alone could leak information. As mentioned above, the paper doesn't seem to actually describe how the noise is scaled, so I can't verify this concern, but I think it's a red flag that the paper does not attempt to reason about this possibility.

3. The paper's Theorem 3.4 comes out of nowhere, its significance is not explained (are users applying it to privatize k of their generalizable points?) and it seems to have been plagiarized from previous work (see "Details of Ethics Concerns" below).

**Summary Of The Paper:**

This paper studies privacy in a federated learning setting. It aims to improve upon methods that directly privatize data points or gradient updates using differential privacy by instead separating data points into "sensitive" and "generalizable" features and then only requiring the client to publish a noisy version of their generalizable features. Each client then constructs their own model based on all shared noisy data and their own raw data. Intuitively, the scale of noise required to privatize the generalizable features may be substantially smaller than that needed to privatize the raw data (as required for local DP), and this enables more accurate final models. The paper concludes with experimental results .

**Summary Of The Review:**

The paper's explanation of its algorithm is unclear, no proofs or code are offered for verifying its reasoning, the privacy guarantee of the algorithm it sketches is suspect even apart from these technical details, and one of the results appears to be plagiarized, with an incorrect proof, from previous work.

---

> ### Author Response · Authors · 2022-11-14
> **Response To Reviewer 52C4**
>
> ___1. This paper studies privacy in a federated learning setting.___
>
> ___A1:___ We appreciate Reviewer 52C4's effort on reading our paper. According to the comments from **Reviewer Kf6q, Lg1w**, and **Yumu**, we focus on defying data heterogeneity in federated learning by sharing information, which is also highlighted in the paper. We will make our paper clearer in the future to avoid such misunderstandings as possible.
>
> __2. I don't understand how the data distillation process works. Is it performed entirely locally? If so, what is the meaning of the E{(x,y)} term?__
>
> ___A2:___ We perform privacy distillation locally, see Sec. 3.3: "The proposed privacy distillation ..." $E_{(x,y)}$ denotes the expectation over the distribution of a client. Notably, privacy distillation is not related to differential privacy.
>
> __3. How can somebody with one data point reason about which parts of it generalize? Do users have multiple data points?__
>
> ___A3:___ The scenario of our work is federated learning. In federated learning, each client naturally has many data points with generalization problems and statistical attributes.
>
>
> __4. the paper doesn't seem actually to describe how the noise is scaled, so I can't verify this concern, but I think it's a red flag that the paper does not attempt to reason about this possibility.__
>
> ___A4:___ For the noise scale, please see Figure 2 (b).
>
> __Plagiarized statement:__
>
> This is not a case of plagiarism.
>   1. Our contribution lies in proposing a new solution to tackle data heterogeneity in federtaed learning, rather than proposing composition theorem.
>   2. We ___did not___ claim we provide a new contribution for differential privacy, i.e., Theorem 3.4, because Theorem 3.4 is a common and standard theorem, which even has become textbook-level knowledge. This is consistent with the comments of **Reviewer Kf6q**: "Theorem 3.4 is the standard adaptive composition result of differential privacy..."
>   3. The original words in our paper are "**utilize composition thoerem**", i.e., Theorem 3.4. Thus, we just use Theorem 3.4 as a tool, because DP is not the main part and only used in a small part of this work.
>
>   The misunderstanding may stem from the fact that some references were mistakenly deleted, such as the definition of DP and Theorem 3.4. Although we don't have plagiarism, the authors here formally apologize to Reviewer 52C4 for any inconvenience incurred.

---

### Official Review · Reviewer_Lg1w · 2022-10-25

**Confidence:** 4
**Correctness:** 2
**Technical Novelty And Significance:** 2
**Empirical Novelty And Significance:** 3
**Recommendation:** 3

**Clarity, Quality, Novelty And Reproducibility:**

The idea is novel conceptually. However, there are missing arguments around the privacy aspect of the proposed method.

**Strength And Weaknesses:**

Strengths:
- The method is novel and the empirical results look promising.

Weaknesses:
- I'm concerned with the privacy guarantee imposed in the work. It doesn't seem that the authors state the clipping bound for Gaussian/Laplacian Mechanism nor does Theorem 3.4 include any information about how the privacy parameter $\varepsilon$ relates to the noise scale $\sigma$ and the clipping bound. That makes me concerned about whether the experimental results comes from unbounded $x_g$? Could the authors explain how sensitivity is bounded in the experiments and theory?
- I'm confused with the separation of $x_s$ and $x_g$ here. Seems that $x_s$ has never been used alone, it seems that the solution is equivalent to generating synthetic data $x_g$ from true data $x$ using a generative model and send the synthetic data to the server. Could the authors explain the motivation of having $x_s+x_g=x$?
- When communicating the synthetic data to the server, how are the labels communicated? It seems that $p(y|x)$ and $p(y|x_g)$ are the same? If that is the case, how to communicate the label information privately?


**Summary Of The Paper:**

This paper proposes a novel method to tackle data heterogeneity in federated learning. Specifically, the authors proposed viewing the data as an addition of public features and private features and generate a set of public features to construct a globally public dataset to aid training. The empirical results demonstrate significant improvement over prior efforts.

**Summary Of The Review:**

Although the experimental results seem good, critical privacy arguments are missing in the current paper (1. bounding sensitivity, 2. communicating label info). Based on current content, justifications and improvements are needed for this work.

**UPDATES**: After reading the responses and other reviewers' comments, I still do not find the privacy argument convincing. The main concerns still lies in unbounded sensitivity and unmasked label info. Therefore, I decide to maintain my initial evaluation.

---

> ### Author Response · Authors · 2022-11-14
> **Response To Reviewer Lg1w**
>
> ___1. How $\epsilon$ relates to $\sigma$ and clipping bound___
>
> *___A1:___*  We guess the clipping bound (the reviewer mentioned) from DP-SGD[CCS16], the typical/seminal work for adding noise to deep learning. Yet, our work does not use DP-SGD, but employs database privacy[1][2][3]. We are sorry that important literature about differential privacy has been accidentally deleted before paper due.
>
>
>
> ___2. The motivation of $x_s + x_g = x$___
>
> ___A2:___  Our method does not synthesize data $x_g$ from true data $x$ using a generative model. In contrast, we predict sensitive features $x_s$ using a generative model, and make $x_g = x - x_s$ generalizable. Accordingly, $x_s$ contains most visual information, which is the reason why we call $x_s$ sensitive features.
>
> ___3. How to communicate the label information privately?___
>
> ___A3:___   Thanks for your valuable question. We communicate the label information without protection. The reasons have two folds: 1) we follow previous work [4,5] to communicate labels; 2) The results of the model inversion attack show that it is hard to attack our method even with exposed label information.
>
> **Reference**
>
> [1] Calibrating noise to sensitivity in private data analysis. Dwork et al. In Theory of cryptography conference, 2006.
>
> [2] Privacy-preserving datamining on vertically partitioned databases. Dwork et al. In Annual International Cryptology Conference, 2004.
>
> [3] Differential privacy for growing databases. Cummings et al. In NeurIPS, 2018.
>
> [4]  No fear of heterogeneity: Classifier calibration for federated learning with non-IID data. Luo et al. In NeurlPS, 2021.
>
> [5] Towards fair federated learning with zero-shot data augmentation. Hao et al. In CVPR, 2021.

---

> > ### Comment · Reviewer_Lg1w · 2022-12-06
> > **Re: Response**
> >
> > Thanks for your response.
> >
> > - To clarify, I'm not mentioning DP-SGD here but in general if you want differential privacy, how do you bound the sensitivity? It seems that the authors simply add random Gaussian noise, leading to unbounded sensitivity. In addition with the unprotected label information, I worried that the proposed method does not provide formal privacy guarantee.

---

### Official Review · Reviewer_Yumu · 2022-10-26

**Confidence:** 3
**Correctness:** 3
**Technical Novelty And Significance:** 3
**Empirical Novelty And Significance:** 3
**Recommendation:** 3

**Clarity, Quality, Novelty And Reproducibility:**

Clarity:
The presentation is in general quite clear except for the privacy analysis and specification in the experiments.

Quality:
Again, my main concern is the privacy analysis.

Novelty:
The general idea seems quite novel.
One minor point: the idea reminds me of https://arxiv.org/pdf/2102.12677.pdf (I might be wrong in understanding either paper though) which decomposes the gradient into a subspace where the majority signal lies in and the orthogonal subspace, and privatized both parts. You might see if there is any connection and anything to borrow there.

Some other comments:
- In "optimization view", the paper mentioned that L in (1) is the cross-entropy loss, but the formalization looks quite general to me that I don't think we need to restrict the loss (or whether the label y should be a scalar or vector).
- The formulations in optimization view and generalization view only focus on the generalizable feature x_g being an accurate approximation to the original data x, but do not contain privacy-related constraints. (The privacy is guaranteed through additive noise.) So it is a bit unclear to me why we should be calling x - x_g the "sensitive features". It looks to me that they are just "ungeneralizable" features and do not necessarily contain sensitive information.

**Strength And Weaknesses:**

Strength:
The paper proposes an interesting idea to distill the generalizable features from samples to improve training.

Weakness:
1. The DP analysis is a bit unclear.
- The paper says that we'll add noise to x_g to form x_p and all the x_p would be sent to the server. That looks like local DP which could usually cause problem to utility. But then the paper says "considering all clients' data as a whole", which seems to suggest something very different from local DP. The authors did not provide a rigorous of sensitivity and privacy analysis.
- It is also unclear whether we're having a per-example privacy or per-user privacy.
- I didn't see the privacy values in the experiments either.
2. The paper claims to separate the generalizable and sensitive features, but it looks more like generalizable vs "non-generalizable" features / the reminder features to me. I don't think the paper has explained much about why the reminder features contain anything sensitive.

**Summary Of The Paper:**

The paper proposes an algorithm that distill and privatize the generalizable features to improve FL training.

**Summary Of The Review:**

The idea seems interesting, but I'm mostly concerned about the privacy analysis.

---

> ### Author Response · Authors · 2022-11-14
> **Response To Reviewer Yumu**
>
> ___1. The authors did not provide a rigorous of sensitivity and privacy analysis.___
>
> *___A1:___* Thanks for the comments. We will make our idea and analysis more clear
>
> ___2. per-samples or per-user___
>
>  *___A2:___* As mentioned in your comments, we add noise on every samples (of shared data) on each local client, so it is per-sample privacy.
>
> ___3. Didn't see the privacy values in experiments___
>
> *___A3:___* Thank you for pointing out this potentially confusing problem. We used the noise scale $\sigma$ in the experiments.
>
>
> ___4. It is a bit unclear to me why we should be calling $x - x_g$ the "sensitive features".___
>
> *___A4:___* In general, the original data $x$ contains sensitive features, so we are prone to keep it locally. In our work, we minimize the information entropy of generalizable features $x_g$, so the remaining part, i.e., $x-x_g$, is almost the same as $x$. That is, $x-x_g$ is sensitive, so we call it sensitive features.

---

### Official Review · Reviewer_Kf6q · 2022-10-28

**Confidence:** 4
**Correctness:** 2
**Technical Novelty And Significance:** 3
**Empirical Novelty And Significance:** 2
**Recommendation:** 3

**Clarity, Quality, Novelty And Reproducibility:**

The paper is broadly well written, but lacks rigor where it is required.
The idea is novel, but the theoretical results (as claimed) are non-existent and empirical improvements are not clear.

**Strength And Weaknesses:**

Strengths:
1. The paper proposes an interesting idea to try and improve federated learning training with differential privacy.
Weaknesses:
1. Theorem 3.4 is the standard adaptive composition result of differential privacy and it is not clearly stated that this is not one of the papers contributions.
2. Theorem 3.5 is an informal statement at best, and in no way shows why the necessary noise to be added is lesser to just the generalizable features. Is the L2 sensitivity lowered in some way?
3. In the experimental evaluations, the high near central training accuracies make it seem like there is some personalization done using the private data on each client $x_s$. Is this the case? If it is the case, the baseline of no local personalization is quite weak. Unfortunately, I couldn't verify this since the code wasn't attached.


**Summary Of The Paper:**

This paper tries to tackle the problem of data heterogeneity in federated learning with differential privacy by attempting to divide the features into private and generalizable features. They ask the question of what is necessary to share to learn global models and can the remaining data stay on the client as local models. They provide empirical evaluation of their claims.

**Summary Of The Review:**

In light of the strengths and weaknesses mentioned above, the weaknesses outweigh the strengths and I recommend the paper for rejection.

---

> ### Author Response · Authors · 2022-11-14
> **Response To Reviewer Kf6q**
>
> ___1. Theorem 3.4 is the standard adaptive composition result of differential privacy, not the paper's contribution.___
>
> *___A1:___*  Thanks for pointing out this confusing problem. The problem stems from the fact that we missed some necessary references related to differential privacy. In response to your valuable comments, we have revised the paper to highlight that: our theoretical analysis is built upon the conclusion in [1][2][3].
>
> ___2. Theorem 3.5 is an informal statement at best, and in no way shows why the necessary noise to be added is lesser to just the generalizable features.___
>
> *___A2:___*  Thanks for pointing out this potentially confusing problem. The sensitive features are kept locally, so we do not need to add noise to non-shared features. Meanwhile, generalizable features are shared globally, so we merely add noise to these features.
>
>
>  ___3. The high near central training accuracies make it seem like there is some personalization done using the private data on each client.___
>
> *___A3:___*     There is no personalization done using sensitive features, i.e., $x_s$. Specifically, for client i, we use its private data and the generalizable features $x_g$ shared across clients to perform local training. Namely, the experimental settings are fair and the experimental improvements are significant.
>
>
> **Reference**
>
> [1] The algorithmic foundations of differential privacy. Dwork et al. In Foundations and Trends® in Theoretical Computer Science, 2014.
>
> [2] The Composition Theorem for Differential Privacy. Kairouz et al. In ICML, 2015.
>
> [3] The Complexity of Differential Privacy. Vadhan. Tutorials on the Foundations of Cryptography, 2017.

---

> > ### Comment · Reviewer_Kf6q · 2022-12-06
> > **Response**
> >
> > Thank you for your response.
> >
> > I am still not convinced why the lesser noise is needed since you are just privatizing the seemingly global features since there is no formal derivation of the sensitivity. I would like to keep my score and recommend rejection for this paper.

---

### Decision · Program_Chairs · 2023-01-20

**Decision:**

Reject

**Justification For Why Not Higher Score:**

The paper lacks rigor. All central claims, including those about privacy, are not supported by formal reasoning.

**Justification For Why Not Lower Score:**

N/A

**Metareview: Summary, Strengths And Weaknesses:**

The paper studies statistical (data) heterogeneity in private federated learning. The proposed idea to address this problem is to divide the features into private and generalizable features. The latter can be extracted from different clients and used in federated training to produce better global models.

Although the idea is natural and interesting, unfortunately, the paper has several major shortcomings.

The paper lacks rigor. There are no formal proofs are given to support the main claims. There is no formal privacy analysis when the paper central claims are about privacy distillation. No discussion is provided on how the noise magnitude is translated to a formal privacy guarantee. Also, the description of the main algorithm lacks clarity. The presentation and writing quality can be improved in several parts of the paper.